# Dynamics of Shrimp Farming in the Southwestern Coastal Districts of Bangladesh Using a Shrimp Yield Dataset (SYD) and Landsat Satellite Archives

**Md Fazlul Karim** [1] , **Xiang Zhang** [1,*] **and Ru Li** [2]

1   School of Resource and Environmental Sciences, Wuhan University, 129 Luoyu Road, Wuhan 430079, China
2   State Key Laboratory of Information Engineering in Surveying, Mapping and Remote Sensing,
    Wuhan University, 129 Luoyu Road, Wuhan 430079, China
*   Correspondence: xiang.zhang@whu.edu.cn

**Abstract:** The shrimp-farming area and shrimp yield are continuously changing in the southwestern coastal districts of Bangladesh. The three southwestern coastal districts, Bagerhat, Satkhira, and Khulna, along with Rampal, a subdistrict of Bagerhat, contribute 75% of the total shrimp yield of Bangladesh. However, the shrimp yield and farming area have declined in Bagerhat district, and the cause of this decline is uncertain. In this research, the differences in the shrimp yield were quantified using a shrimp yield dataset (SYD) and k-means classification. A supervised image classification approach was applied to quantify the spatiotemporal changes and identify the influencing factors behind the declining shrimp-farming area and yield in Rampal, Bagerhat district, using Landsat satellite archives. K-means classification reveals that, between 2015 and 2017, the shrimp yield in Bagerhat district declined significantly compared to Satkhira and Khulna. The satellite-based monitoring results affirm that the shrimp-farming area of Rampal also decreased rapidly, from 21.82% in 2013 to 6.52% in 2018. This research estimates that approximately 70% of the shrimp-farming area was lost in Rampal since December 2013. Hence, the findings of this research might motivate the responsible bodies to declare the shrimp-farming coastal area as a "shrimp zone" and implement an active policy to protect the vulnerable shrimp-farming industry and shrimp farmers, considering it is the second-largest export earning source in Bangladesh after ready-made garments.

**Keywords:** temporal shrimp yield; Landsat; k-means classification; supervised image classification; Rampal thermal power plant; spatiotemporal change analysis

## 1. Introduction

The shrimp yield and farming area in the southwestern coastal districts of Bangladesh have been dynamically regulated over the years; the region has ideal climatic conditions and the industry has good labor costs [1,2]. Shrimp is the second largest export product in Bangladesh after ready-made garment commodities (e.g., garment products, textile items, and vegetable textiles/yarns) [3] and has already become a multimillion-dollar industry [4]. Three districts, Bagerhat, Satkhira, and Khulna, along with Rampal, a subdistrict of Bagerhat, are the significant coastal shrimp-farming districts of Bangladesh, making a major contribution to the national economy over the past two decades [5]. These three southwestern districts contributed 75% of the total shrimp industry between 2002 and 2017 [6–21].

The shrimp yield of these southwestern coastal districts has changed continuously since the commencement of profit-oriented business in 1970 [22]. Ahmed and Diana (2015) assessed the impact of different climatic variables on shrimp farming [23]. Ali (2006) investigated the impact

of shrimp farming on rice production, aquatic habitats, and soil properties [24]. Afroz and Alam (2012) addressed the severe impacts of uncontrolled shrimp farming [25]. Ahmed (2013) reviewed the issues key to meeting environmental, social, and economic challenges through prawn and shrimp farming [26]. Alam et al. (2007) explored the costs and returns of shrimp farming in disease-affected areas [27]. Matin et al. (2016) evaluated the present shrimp-farming situation in the southwestern coastal districts [2]. To the best of our knowledge, at the time of this research, little research has quantified shrimp yield changes utilizing focused group discussions, questionnaire surveys, and informant interviews. In order to assess shrimp yield changes between 1995 and 2015 from a historical perspective, Akber et al. (2017) employed a systematic random sampling method and stated that the shrimp yield is declining in the selected study area [22]. Akber et al. (2017) conducted research considering only six subdistricts of the southwestern coastal districts of Bangladesh; it is controversial that this study did not address the actual differences in shrimp yield of all southwestern coastal districts.

In order to quantify the shrimp yield differences and address the overall pattern of shrimp yield changes, the shrimp yield dataset (SYD) and k-means classification were used in this study. K-means clustering [27], hierarchical clustering [28], and Gaussian mixture model clustering [29] are some of the standard clustering classification methods for change analysis. Agarwal et al. (2013) used k-means classification for a crime analysis of England and Wales from 1990 to 2011 to specify crime trends [30]. Marino et al. (2018) implemented a k-means algorithm to monitor the evolution of the academic performance of students in a higher educational institution of Nigeria for academic planners to make adequate decisions [31]. For sequence plan optimization in steel production, Svecet et al. (2016) used k-means classification in order to achieve agreement between capabilities of production and order requirements in a given period [32]. To determine the water quality authentically and effectively, Zou et al. (2015) used a varying weights k-means classification technique adopted for water quality analysis of the Heihe River in China [33].

It is worth noting that about 80–90% of livelihoods in the southwestern coastal districts of Bangladesh depend on shrimp farming [2]. However, the shrimp-farming area at Rampal, Bagerhat district, has changed a great deal over the past two decades along with the shrimp yield, as evidenced by government-published Fisheries Resource Survey System (FRSS) reports, newspaper articles, and so on [6–21]. According to the FRSS data and existing research, shrimp production and the shrimp-farming area of Bagerhat district have been declining compared to Satkhira and Khulna districts in recent years [34], the cause of which is uncertain and politically contentious. Akber et al. (2017) stated that the outbreak of disease at shrimp farms, low shrimp prices, and high labor costs accounted for the decline in shrimp-farming area and yield [22]. Ali et al. (2006) affirmed that long-term environmental consequences such as increased salinity and a loss of biodiversity were equally responsible for the decline in shrimp yield and farming area in the southwestern coastal districts of Bangladesh [24]. Ahmed and Diana (2015) stated that climatic variables such as cyclones, coastal flooding, drought, sea-level rise, and sea surface temperature have severe negative impacts on the production and growth of shrimp [23]. Apart from the above factors, various researchers, local people, and shrimp farmers have pointed out that the 1320 MW coal-fired thermal power plant in Rampal appears to be a primary cause of the declining shrimp-farming area and yield since 2013.

On 2 January 2012, two years before the Environmental Impact Assessment (EIA) was approved, the Bangladeshi government handed over 1834 acres of land in Rampal to the Bangladesh Power Development Board (BPDB) in order to boost the power production of the country. Only 86 acres of this procurement land was state-owned; the rest was privately inherited shrimp farming and agricultural land [35]. Since construction work on the Rampal thermal power plant began in April 2017, it has led to the destruction of livelihood options (e.g., shrimp farming and agriculture) for local communities [36]. Landless farmers, environmentalists, nongovernment organizations, and residents of the Rampal region protested against the setting up of the power plant well before a Memorandum of Understanding (MoU) was signed between the National Thermal Power Corporation of India and

BPDB on 1 November 2010 [35]. Organizations such as Greenpeace and Water-Aid and residents of both Bangladesh and India pointed out, that aside from the fact that many shrimp farmers and agricultural landlords had already become landless [37], the coal-based power plant would lead to severe public health emergencies in the surrounding area due to harmful health effects soon after the power plant became operational [38]. The prospect of cheap power from Rampal has already attracted many industries, all of which are operating within a 10-km radius of the power plant site, which was previously used for shrimp farming [39]. Chowdhury (2017) asserted that mitigating the shrimp farm loss would be very difficult in Rampal [40]. To the best of our knowledge, no land use/cover change analysis has been done in Rampal using remote sensing technology at the time of this research, although this area has economic importance for the development of the entire country.

Land cover changes occur with the transformation and conversion of different land cover types and show the existing complex interactions between humans and the physical environment [41]. A combination of remote sensing (RS) data and geographic information system (GIS) techniques helps us to study land cover changes at low cost and in less time [42]. Maximum likelihood (ML) [43–45], random forest (RF) [46], decision tree (DT) [47], support vector machine (SVM) [48], and neural network (NN) [49] classifiers are some of the conventional supervised land cover classification methods. Widely used image classification methods, such as ML [43,50–55], work on a uniscale pixel-by-pixel basis and ignore multiscale information within the image and spatial information surrounding the pixels [56]. ML classification techniques often fail to differentiate between different forest types, agriculture, and grassland [57]. Change detection is one of the foremost topics in land cover monitoring [58] and is useful in regional planning and policy making [59].

The ultimate goal of this study is to quantify the differences in temporal shrimp yield in the three southwestern coastal districts of Bangladesh, as well as identify the critical factors behind the rise and fall of the shrimp-farming area in Rampal, Bagerhat district, utilizing machine learning, RS, and GIS techniques. The present study introduces a new approach to quantifying differences in shrimp yield in these districts. This paper also presents a new tool for evaluating aspects of the aquaculture industry and land use planning. This study will be helpful for the bodies responsible for the development, planning, and policymaking of the shrimp sector, the second-largest export earning source of Bangladesh. This study implemented RS and GIS techniques in order to assess the temporal changes in shrimp-farming areas utilizing supervised image classification that will undoubtedly save time and cost over traditional methods for the responsible government organizations. Furthermore, the analysis of this research can be a source of guidance for decision-makers, planners, and development partners who intend to work for the advancement of the shrimp-farming sector. The analysis of this study will also be beneficial to readers, because it will add to the existing literature in this particular area of interest, given that few scholars have written about this topic.

The objectives of this research are twofold: first, to quantify the differences in shrimp yield of the three major shrimp-farming districts of southwestern Bangladesh based on the SYD from 2002 to 2017 in order to understand the changing patterns of shrimp yield; and second, to characterize and map temporal changes of the shrimp-farming area and identify influencing factors behind the declining shrimp-farming area in Rampal, Bagerhat district, in order to support long-term monitoring and evaluation of shrimp farming in the coastal districts of Bangladesh.

## 2. Materials and Methods

### 2.1. Study Area

The study area is Rampal, a prominent shrimp-farming Upazila (subdistrict) of Bagerhat district, located in the southwestern part of Bangladesh, as shown in Figure 1c, ranging between 22°34′ N and 22°39′30″ N latitude and 89°35′30″ E and 89°46′30″ E longitude.

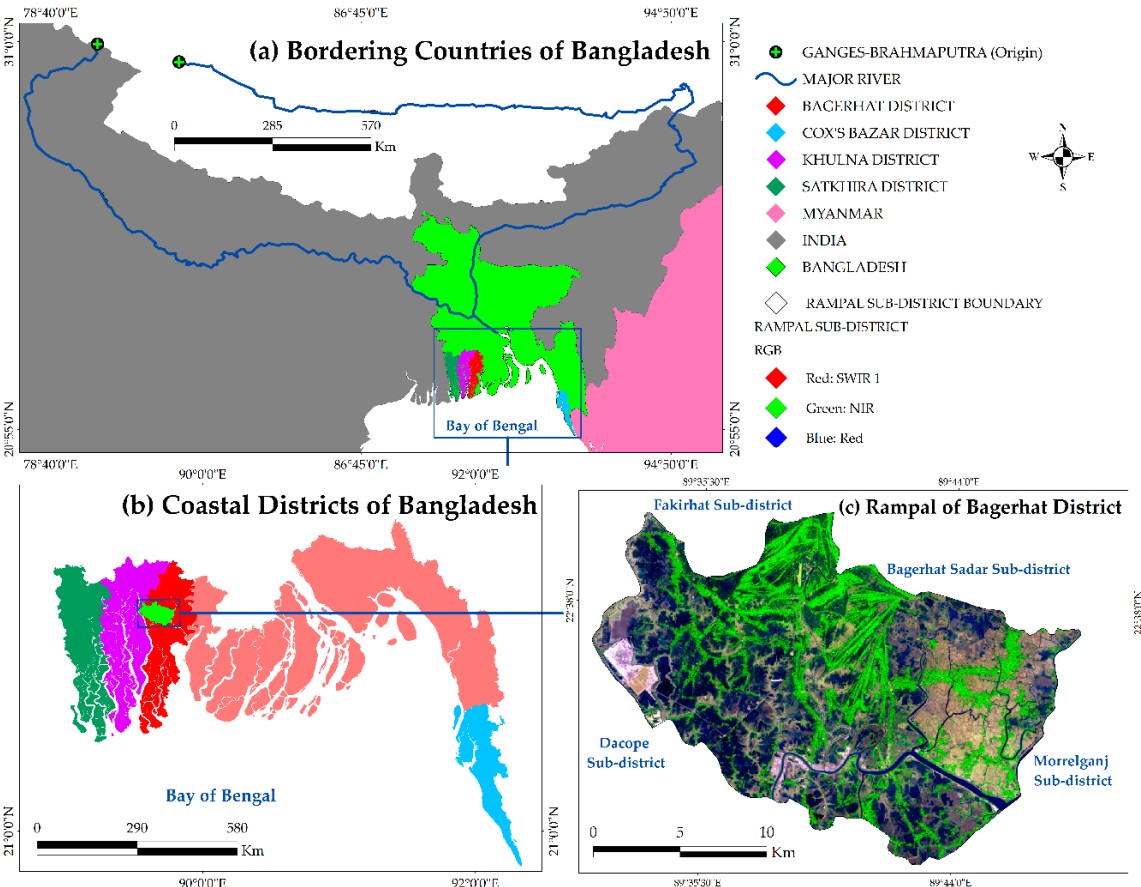

**Figure 1.** Schematic diagram of the study area, Rampal. (**a**) Bangladesh bordered by India, Myanmar, and the Bay of Bengal; (**b**) the significant shrimp-farming coastal districts of Bangladesh—these are the focus of our analysis of quantifying temporal differences in shrimp yield; (**c**) false-color composite image of the study area, Rampal with a band combination of 6, 5, and 4.

Rampal is one of the prime shrimp-farming subdistricts of Bangladesh, considering its contribution to the national economy. Rampal has a dynamic river system with three large rivers: the Passur, Poylahar, and Khulirjar. Rampal has 149 villages grouped into 11 clusters (called unions) and covers 326.18 square km. Rampal is about 12 km from the Mongla port, the second international seaport of Bangladesh, 10 km from the Mongla Export Processing Zone (EPZ), and 25 km from the world's largest mangrove forest, Sundarban (see Figure 2a).

The study area, Rampal, is situated in the subtropical monsoon climatic zone, with wide seasonal variations in high temperature, rainfall, and humidity. There are three distinct seasons: a hot, humid summer (March–June), a cool, rainy monsoon (June–October), and a dry winter (October–March). The monthly average dry bulb temperature in this area from 1991 to 2016 was 34.23 °C/month. The annual average rainfall from 1989 to 2016 was 1908.21 mm/year. Most of the soil in this area is clay, and the average elevation of the study area is 2 m (7 ft). A considerable amount of brackish water, the subtropical climate, and low labor costs provide a unique opportunity for shrimp farming in Rampal [60]. Rampal is locally known as the "Kuwait of Bangladesh" due to its sizeable production and high revenue in shrimp farming [61], analogous to oil as the main export of Kuwait. Shrimp is also recognized as "white gold" due to the amount earned per metric ton [23].

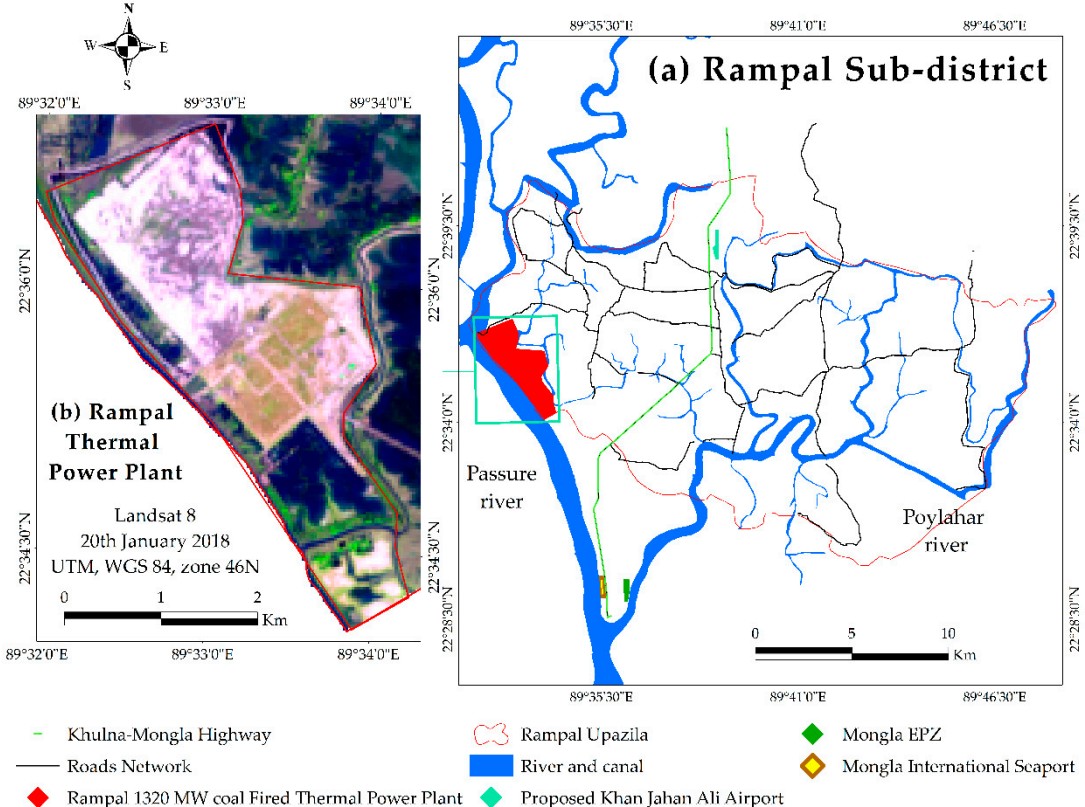

**Figure 2.** (**a**) Geographic setting of Rampal; (**b**) false-color composite image of the 1320 MW coal-fired thermal power plant site and its surrounding area with the band combination of 6, 5, and 4.

However, the shrimp-farming area in Rampal has been declining in recent years. Various reports have suggested that the thermal power plant has been a primary cause of the decline, along with other significant factors, such as the impacts of climatic variables, recent low shrimp prices, high labor cost, and disease outbreak, being equally responsible. The Rampal thermal power plant project site lies between 22°37′0″ N and 22°34′30″ N latitude and 89°32′ E and 89°34′5″ E longitude [35,40], as shown in Figure 2b.

Bagerhat, Khulna, and Satkhira, along with Rampal, a subdistrict of Bagerhat, are the top shrimp-producing districts of Bangladesh, and the shrimp yield and farming area have declined more in the Bagerhat district than in the other two districts. The data collection, selection criteria, and satellite image processing are discussed in the next section.

## 2.2. Data Collection and Satellite Image Processing

Diverse datasets, including geospatial, demographic, and satellite data (shown in Table 1), were utilized to achieve the objectives of this research. The shrimp yield dataset (SYD) was created from the FRSS yearbook of Bangladesh (https://document.bdfish.org/) in order to quantify the temporal differences in shrimp yield of the three southwestern coastal districts between 2002 and 2017 [6–21]. The FRSS yearbook provides annual district-level fishery production and area information; therefore, it is worthwhile for national, regional, and global aquaculture development planning as well as change analysis. However, the FRSS yearbook does not provide information on subdistrict or lower-level fishery production, farmed areas, or fishery yield. Also, district-wise shrimp production data of Bangladesh before 2002 were unavailable, and the data for 2018 had not yet been published at the time of this research.

**Table 1.** Descriptions of shrimp yield dataset, geospatial, and satellite data. TM, Thematic mapper; OLI, Operational Land Imager; TIRS, Thermal Infrared Sensor; USGS, U.S. Geological Survey.

| Data | Acquired Date/Year | Producer |
|---|---|---|
| Landsat 5 (TM) | 19 January 2000<br>22 January 2007 | Downloaded from USGS global land cover facilities<br>(http://glovis.usgs.gov/) |
| Landsat 8 (OLI/TIRS) | 24 December 2013<br>20 January 2018 | |
| Geospatial data (administrative boundary of Bangladesh, Rampal, southwestern coastal districts) | 2018 | Humanitarian Data Exchange (HDX) (https://data.humdata.org/) |
| Google Earth historical images | 31 December 1999<br>13 April 2008<br>15 January 2013<br>8 January 2017 | Digital Globe |
| Shrimp yield dataset (SYD) | 2018, created from Fisheries Resource Survey System (FRSS) data of Bangladesh | (https://document.bdfish.org/) |

In this research, to detect changes in the shrimp-farming area and identify the influencing factors behind the declining shrimp-farming area in Rampal between 2000 and 2018, we acquired multispectral Landsat satellite images for four time periods. The image scenes were taken by the U.S. Geological Survey (USGS) at 30 m ground resolution. To acquire cloud-free images of Rampal, we selected satellite images from the winter (December–February), when cloud cover was around 0–10%. Out of 21 available images, we selected four satellite scenes for maximum likelihood (ML) supervised image classification after visually inspecting all of the downloaded images.

ENVI 5.3 was used for atmospheric correction by conducting radiometric calibration, Fast Line-of-sight Atmospheric Analysis of Hypercubes (FLAASH), and Band Math. The minimum noise fraction (MNF) forward transform wizard was used to determine the authentic dimensionality of image data to segregate noise. All satellite images were registered in the same projection system (UTM, WGS 84, zone 46N) to avoid spatial reference problems. We used Google Earth historical images of 31 December 1999, 13 April 2008, 15 January 2013, and 8 January 2017, and photo-interpretation of the downloaded Landsat images for tracing previously and newly settled shrimp-farming areas, identifying land cover features, creating training samples, and conducting an accuracy assessment of the classified images.

*2.3. Shrimp Yield Dataset*

In sequence to calculate the hectare-wise shrimp yield, we created the shrimp yield dataset (SYD). The equation used to calculate the SYD is as follows:

$$\text{Shrimp Yield Dataset (SYD)} = \frac{\text{Production (Mt)}}{\text{Farmed Area (Ha)}}. \tag{1}$$

We created the SYD to quantify the annual differences in per-hectare shrimp yield in the three southwestern coastal districts (Bagerhat, Satkhira, and Khulna) between 2002 and 2017. We created the SYD based on the FRSS yearbook (2002–2017) of Bangladesh. In the next section, we briefly discuss the procedure to quantify the shrimp yield changes utilizing k-means classification.

*2.4. Shrimp Yield Difference Quantification Using K-Means Classification*

The k-means classification approach was used to quantify the temporal differences in shrimp yield and understand the changing patterns of shrimp yield. K-means classification allows for the

identification of observations that are alike, and potentially categorizes them [30] and their clustering quality [33,62,63]. The analysis was run through R studio and associated packages.

The distance measure is a critical step in clustering. It defines how the similarity of two yield observations ($x_i$, $x_j$) is calculated and influences the shape of the clusters. In this research, in the SYD there are 16 yield observations (the 16 years from 2002 to 2017, represented as x vector) with three variables (yields of Bagerhat, Satkhira, and Khulna). In order to measure the distance, we used the Euclidean distance algorithm. The Euclidean distance was computed to represent the dissimilarity between each pair of observations, and the equation is as follows:

$$Dis\left(x_i,\, x_j\right) = \sqrt[2]{\left\{\left(x_i^1 - x_j^1\right)^2 + \left(x_i^2 - x_j^2\right)^2 + \left(x_i^3 - x_j^3\right)^2\right\}},\tag{2}$$

where $x_i$ is the yield observation with variables $x_i^1$, $x_i^2$, and $x_i^3$, representing the yield of the three districts. There are 16 observations in total in the research between 2002 and 2017, where ($x_i$, $x_j$) ($i = 1$, $2, \ldots, 16$; $j = 1, 2, \ldots, 16$) represents a pair of yield observations. $Dis\left(x_i,\, x_j\right) = 0$ whenever $i = j$.

The primary step when using k-means clustering is to indicate the number of clusters (k) that will be generated in the final solution [64]. There are three popular methods for determining the optimal clusters: elbow method, silhouette method, and gap statistics [29]. The most popular among researchers is the elbow method [64], which is also used in this research. We first plotted the curve of total within-cluster sum of squares (WSS) according to the number of clusters k (in this case k = 10). Generally, the location of a bend (elbow) in the plot is considered as an indicator of the appropriate number of clusters [62]. We used the standard k-means algorithm, the Hartigan-Wong algorithm, which has several potential advantages compared to the classical and prevalent optimization heuristic known as Lloyd's algorithm [65].

## 2.5. Image Classification

The goal of this study was to identify the influencing factors behind the declining shrimp-farming area; we adopted a simple classification system, partly derived from Anderson et al.'s (1976) [66] first-order hierarchical classification system. A variety of land use/land cover (LULC) types were observed in Rampal, including forest, homestead vegetation, agricultural land, construction and open areas, water bodies, shrimp farms, bare land, and others. Initially, five representative land cover categories (vegetation, water bodies, shrimp farms, bare land, and cultivated land) were generated using expert knowledge of the study area and observations from a field survey undertaken in July 2016, presented in Table 2.

**Table 2.** Land use/land cover (LULC) classes delineated based on supervised classification (based on [43,50–53]).

| N | Land Cover Type | Description |
|---|---|---|
| 1 | Vegetation | Mixed forest land, scattered forest, sparse low-density forest, degraded forest, mix of trees and other natural grass cover, homestead vegetation |
| 2 | Shrimp farms | Blackish water containing saline |
| 3 | Water bodies | Rivers, permanent open water, canals, ponds, reservoirs |
| 4 | Bare land | Areas of exposed soil, roads, isolated and clustered settlements, barren areas influenced by human impact |
| 5 | Cultivated land | Paddy fields, wet and dry crop fields, fallow lands |

The Google Earth historical images of 31 December 1999, 13 April 2008, 15 January 2013, and 8 January 2017, photo interpretation (four downloaded Landsat images of 2000, 2007, 2013, and 2018), and field visit were used to recognize and confirm the different features [42]. The training samples were generated using polygon vectors for each feature based on their spectral reflectance

wavelengths, shown in Figure 3. The polygons were drawn manually around the selected features to create regions of interest (ROIs). Confusion among the features was avoided by having satisfactory spectral reflectance wavelengths for each class [43].

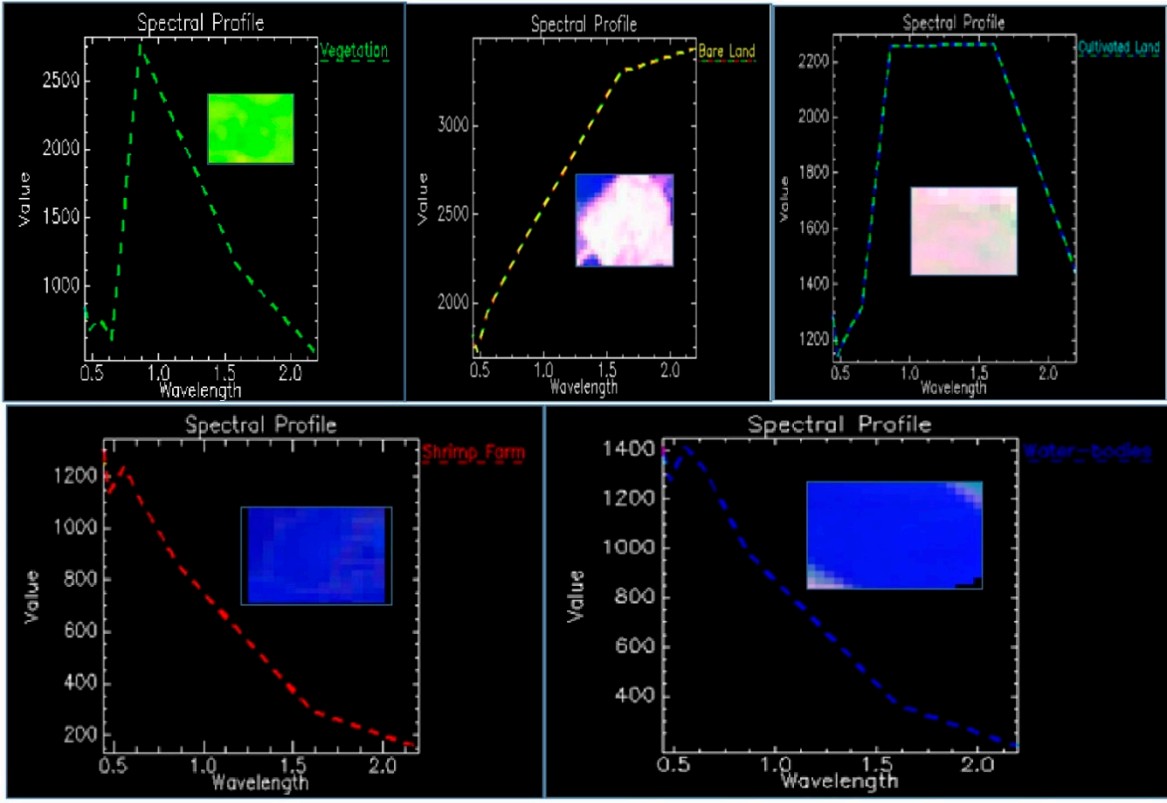

**Figure 3.** Spectral reflectance wavelength of selected training features. Colors represent mean reflectance wavelength values: green = vegetation; blue = water bodies; red = shrimp; yellow = bare land; and cyan = cultivated land.

Repeated segmentation and testing of previews of each land cover/use class helped to improve the accuracy [67]. The same band set—shortwave infrared-1 (SWIR-1), near infrared (NIR), and red (bands 6, 5, 4, and bands 5, 4, 3 for Landsat-8 Operational Land Imager (OLI) and Landsat-5 Thematic Mapper (TM), respectively)—were used for maximum likelihood classification [68,69], in order to minimize the bias caused by using different band combinations. The LULC transition areas and their percentages for classified images from 2000, 2007, 2013, and 2018 were subsequently derived from the classification results using ENVI 5.3. The change detection process of classified images can be performed in several ways: image differencing, thematic change workflow, or change detection difference mapping [70–72]. In this paper, the thematic change workflow process was carried out to detect 2000–2007, 2007–2013, 2013–2018, and 2000–2018 land use/cover differences. Transitional probability matrices utilizing post-classification change detection statistics function were used to indicate the conversion of land use and land cover from one category to another [73]. The sequential explanation of the whole research process is described in Figure A1.

## 3. Results

### 3.1. Temporal Differences in Shrimp Yield from 2002 to 2017 Using K-Means Classification

The annual shrimp yield between 2002 and 2017 is plotted in Figure 4, where the differences in yield before and after 2009 are noticeable. However, to understand the changing pattern of shrimp yield

from a quantitative perspective, we used k-means classification, aiming to divide the yield observation data into cluster groups.

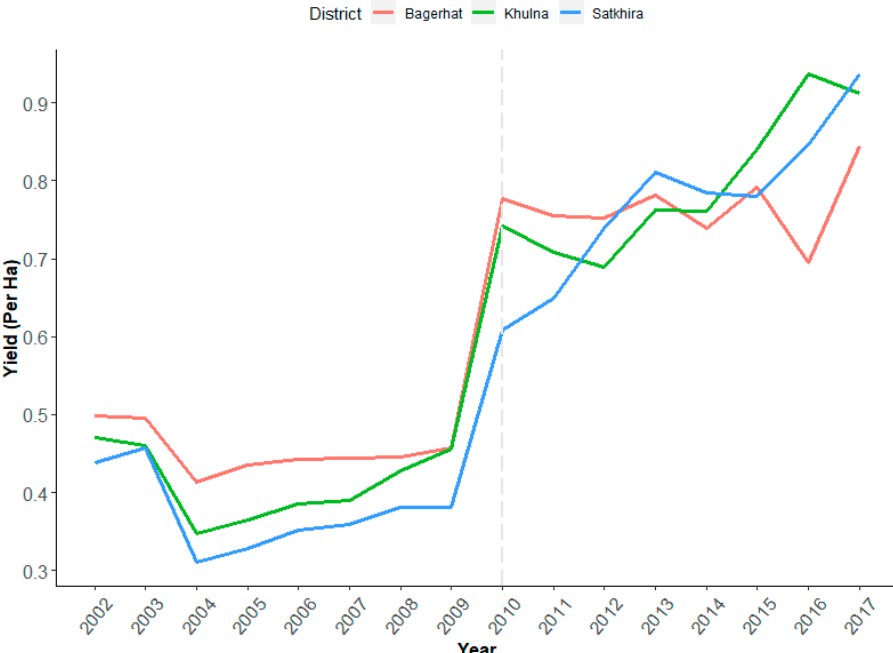

**Figure 4.** yield distribution (per hectare) of the three significant shrimp-farming coastal districts of Bangladesh between 2002 and 2017.

To represent the dissimilarity between pairs of observations, in this research, we computed the Euclidean distance. The normalized dissimilarity result between each yield observation is demonstrated in Figure 5. Higher values represent more substantial dissimilarity. Three types of subgroups were found to be similar and visible in the heat map, as follows: the cyan area of 2002–2009, the light red area of 2010–2015, and the relatively dark red area of 2016–2017.

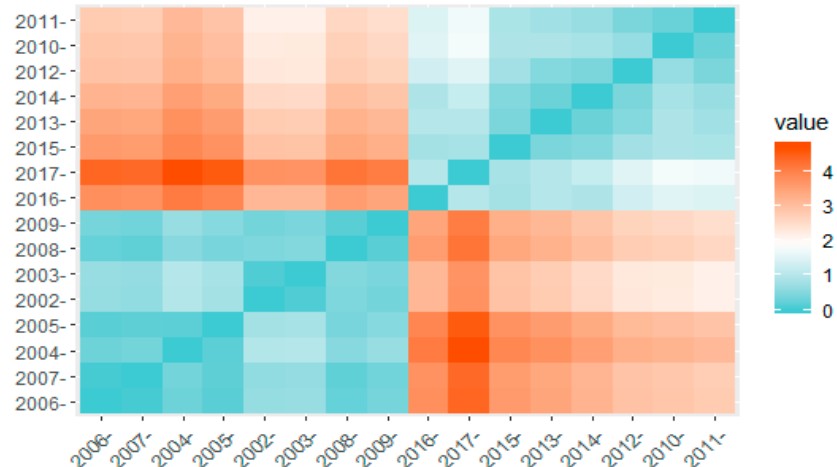

**Figure 5.** Heat map of SYD dissimilarity, by year, from 2002–2017. High values represent more substantial dissimilarity, and low values represent datasets that appear to be reasonably similar.

In this research, we used the elbow method to determine the optimal cluster number. In this case, the optimal number of clusters is three, as shown in Figure 6.

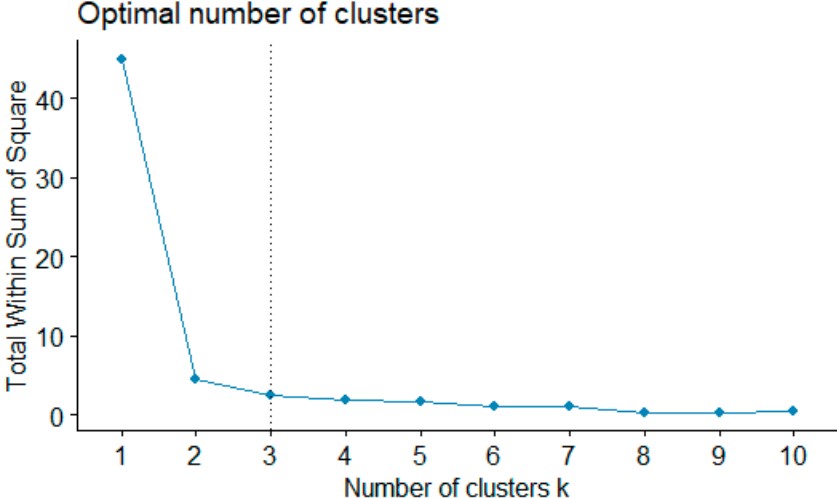

**Figure 6.** Deciding the optimal number of clusters for yield cluster analysis. The optimal number is three in this case, as it appears to be the bend in the elbow; in addition, from Figure 5, it is evident that our data can be divided into three subgroups.

Since the number of clusters must be set before running the algorithm, it is often advantageous to use several values of k and examine the differences in the result [63]. However, in this research, we used different values of k (2, 3, 4, and 5) before setting the final number of clusters as three. The k-means classification results can be seen in Figure 7.

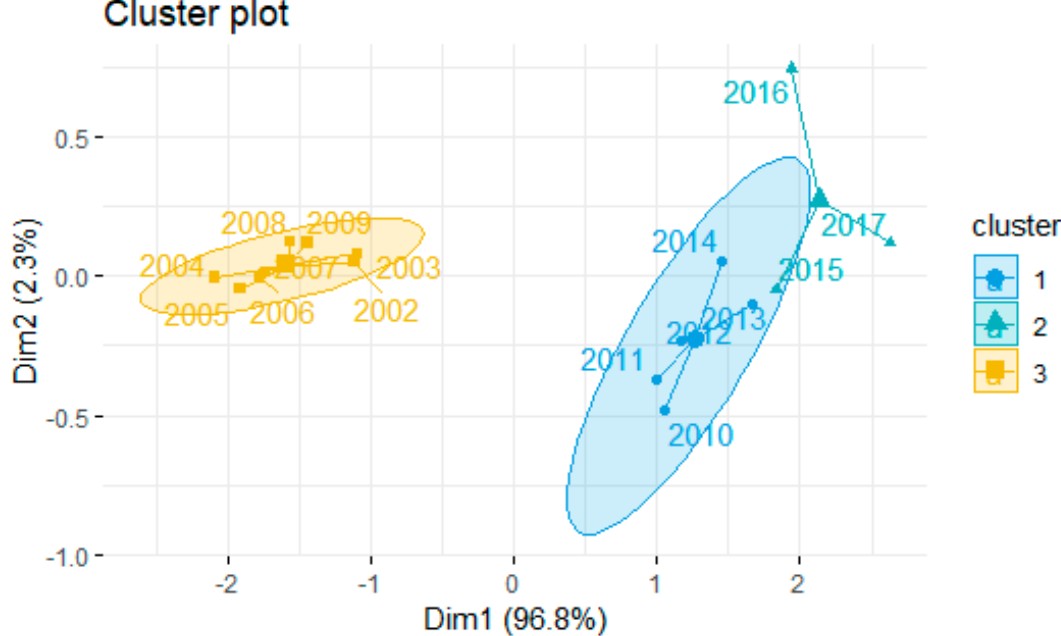

**Figure 7.** Clustering results of differences in shrimp yield between 2002 and 2017 in the southwestern coastal districts of Bangladesh. The results show the shrimp yield dataset grouped into three clusters. dim1 (96.8%) and dim2 (2.3%) explain that 96.8% of the variance within the data is captured by principal component analysis 1 (PCA 1), while PCA 2 describes 2.3% of the variance of the dataset.

Figure 7 and Table 3 show the shrimp yield differences over the study period. The clusters of the yield dataset are similar to each other and different from others [74]. The result indicates that the years between 2002 and 2009 belong to cluster 3. The average mean of cluster 3 is −0.92, suggesting that the differences of shrimp yield in the three southwestern coastal districts of Bangladesh

were small. Similarly, from 2010–2014, included in cluster 1, the average mean is 0.73, suggesting that the differences were moderate. Likewise, from 2015–2018, linked with cluster 2, the average mean is 1.23, affirming large differences of shrimp yield in the three districts.

**Table 3.** K-means classification results of SYD, clustering vector groups, mean values, and average mean values; high mean value represents large difference in shrimp yield.

| Cluster | Bagerhat (Mean) | Satkhira (Mean) | Khulna (Mean) | Average Mean | Clustering Vector |
|---|---|---|---|---|---|
| 1 | 0.91 | 0.67 | 0.62 | 0.73 | 2010, 2011, 2012, 2013, 2014, |
| 2 | 1.00 | 1.29 | 1.40 | 1.23 | 2015, 2016, 2017 |
| 3 | −0.95 | −0.90 | −0.91 | −0.92 | 2002, 2003, 2004, 2005, 2006, 2007, 2008, 2009 |

Moreover, the result also demonstrates that from 2002 to 2009, the shrimp yields of the three significant shrimp-farming coastal districts were very similar (close to −0.9). From 2010 to 2014, the yield of Khulna district was the lowest (0.62) and that of Bagerhat was the highest (0.91). However, from 2015 to 2017, Khulna had the highest (1.40) and Bagerhat had the lowest (1.00).

### 3.2. Image Classification and Accuracy Assessment Results

The confusion matrix method [53] using ground truth ROI for each of the five classes was applied by segregating test pixels to the corresponding location in the classified images to assess the accuracy of the classified images of 2000, 2007, 2013, and 2018. In this study, the training area/reference data (223 polygons and 8618 pixels for two Landsat TM images and 207 polygons and 14,883 pixels for two Landsat OLI images) were randomly and manually selected to assess classification accuracy [44,52]. The producer and user accuracy were obtained through a confusion matrix [44,45]. Overall classification accuracy for 2000, 2007, 2013, and 2018 is 99.22%, 97.80%, 90.23%, and 92.03%, with kappa coefficient index values of 0.98, 0.97, 0.88, and 0.89, respectively.

Nowadays, improved image classification algorithms and satellite imagery with excellent resolution make it easy for researchers to conduct data mining and monitoring of a broad range of target features on the ground. ML classification has become a popular and efficient technique for RS applications, especially for land use/cover monitoring. ML classification has been used extensively for different research objectives in the recent past, due to its ability to deal with complex relations among variables coupled with high classification accuracy [75].

Classification results of the classified images of Rampal in 2000, 2007, 2013, and 2018 are shown in Figure 8. According to these results, the statistics of different types of land use areas and their proportions are shown in Figure 9. By visual interpretation, it is easy to state that the shrimp farm, bare land, and cultivated land classes of the study area have a dynamic (transition and conversion) relationship over the years. The shrimp-farming area in Rampal experienced the most change in the study area over the years. According to the classification results, the shrimp farm area increased notably in the first 13 years and then decreased rapidly in 2018: the area was 2246.67, 3593.07, 7117.47, and 2128.14 hectares in 2000, 2007, 2013, and 2018, respectively.

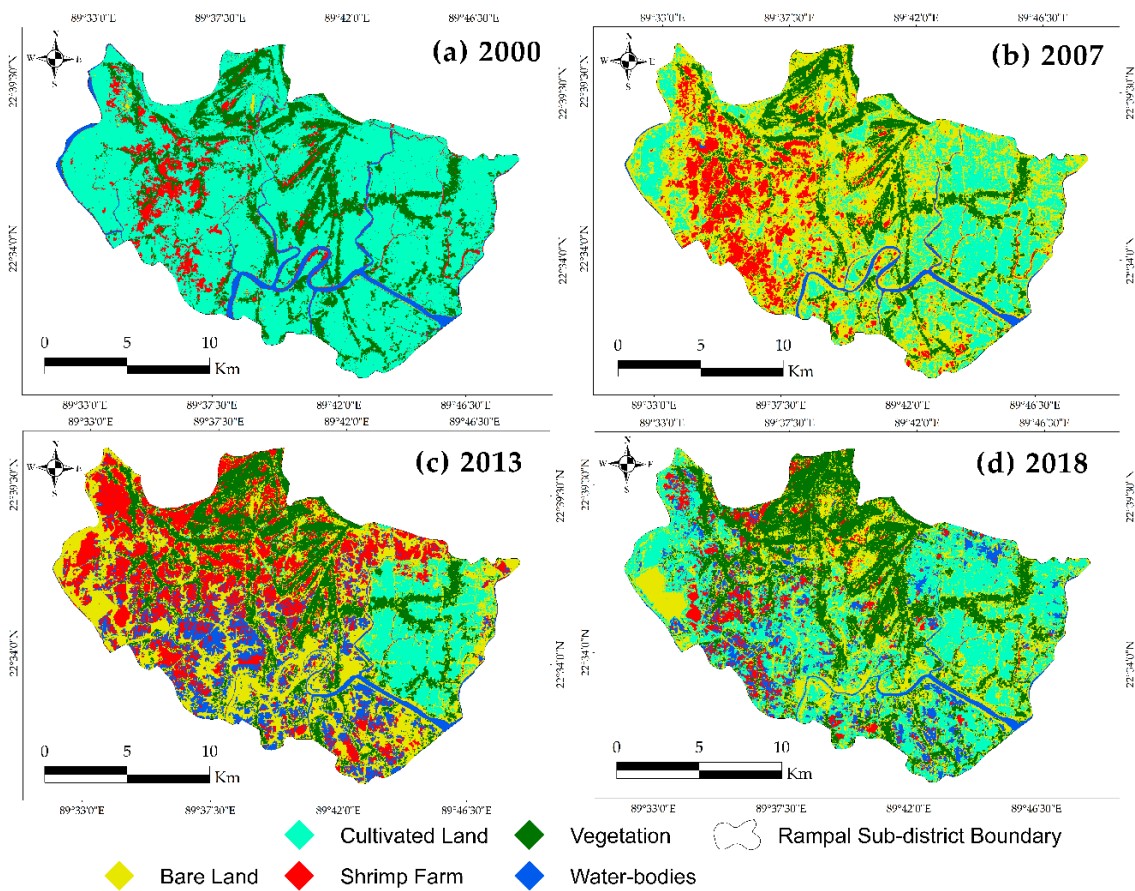

**Figure 8.** Land cover maps for the study area, classified into five major land cover classes: bare land (yellow), cultivated land (cyan), shrimp farms (red), vegetation (green), and water bodies (blue); at four time periods: (**a**) 2000, (**b**) 2007, (**c**) 2013, and (**d**) 2018.

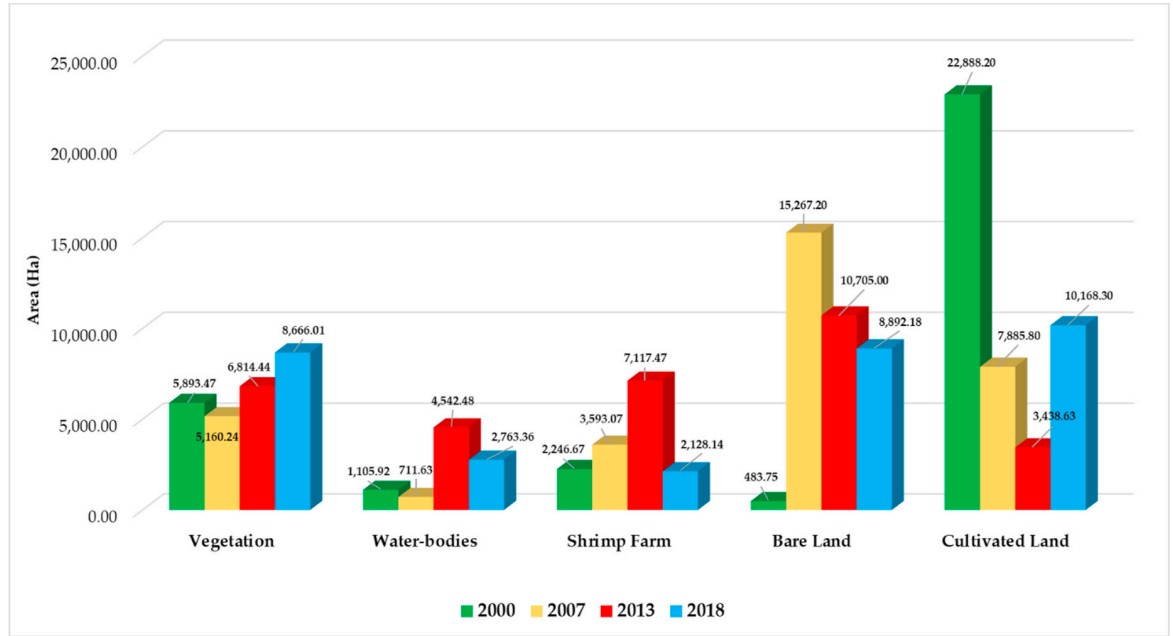

**Figure 9.** Changes in LULC areas (in hectares) in Rampal, 2000–2018.

Table 4 illustrates the differences in net land use of each class in 2000–2007, 2007–2013, 2013–2018, and 2000–2018. Dynamic land-use changes were observed in Rampal over the study period. The most pronounced changes were a rapid increase in shrimp-farming area (1346 hectares, 59.93%) and a noticeable increase of bare land (14,783 hectares, 3056.02%) as well as an associated decrease of cultivated land cover (15,002 hectares, 65.55%) in 2000–2007. There was a rapid increase of shrimp-farming area (3524 hectares, 98.09%) and decrease of bare land area (4562 hectares, 29.88%) and cultivated land (4447 hectares, 56.40%) in 2007–2013. There was a brisk and sudden decrease of shrimp farm area (4989 hectares, 70.1%) and increase of cultivated land area (6730 hectares, 195.71%) in 2013–2018.

**Table 4.** Gain and loss of different LULC areas (in hectares and percentage) in Rampal, 2000–2018.

| LULC Class | 2000–2007 | | 2007–2013 | | 2013–2018 | | 2000–2018 | |
|---|---|---|---|---|---|---|---|---|
| | **Area** | **%** | **Area** | **%** | **Area** | **%** | **Area** | **%** |
| Vegetation | −733 | −12.44 | 1654 | 32.06 | 1852 | 27.17 | 2773 | 47.04 |
| Water bodies | −394 | −35.65 | 3831 | 538.32 | −1779 | −39.17 | 1657 | 149.87 |
| Shrimp farms | 1346 | 59.93 | 3524 | 98.09 | −4989 | −70.1 | −119 | −5.28 |
| Bare land | 14,783 | 3056.02 | −4562 | −29.88 | −1813 | −16.93 | 8408 | 1738.18 |
| Cultivated land | −15,002 | −65.55 | −4447 | −56.40 | 6730 | 195.71 | −12,720 | −55.57 |

*3.3. Factors behind the Rise and Fall of Shrimp Farming in Rampal*

3.3.1. Declining Shrimp-Farming Area and Increased Bare Land in Rampal, 2000–2018

The shrimp-farming area in Rampal experienced a harmonious relationship between the growth of bare land and loss of cultivated land from 2000–2018. The shrimp-farming area increased gradually in the first 13 years, between 2000 and 2013, accounting for 6.89% to 21.82%. However, it decreased rapidly to 6.52% in 2018 due to increased anthropogenic activities followed by a growth of bare land from 2000 to 2018, accounting for 1.48% to 27.26%. Figure 10 shows a schematic map illustrating the gradual increase and sudden rapid decrease of shrimp-farming area in Rampal between 2000 and 2018. Figures 11 and 12 show a vector layer sketch map illustrating the conversion from other land covers to shrimp farm (gain) and from shrimp farm to other land covers (loss). It affirms that cultivated land was the prime source of shrimp-farming area, shown in sky blue in Figure 11a.

In 2000, there were 2246.67 hectares used for shrimp farming, and in 2007 there were 3593.07 hectares. Results confirm that out of the 3593.07 hectares, a vast amount of the shrimp-farming area, 2220 hectares (9.70%), was converted from cultivated land, as shown in Figure 11a. Results also show that 861 hectares (38.32%) of shrimp farm were converted to bare land in the same time period, represented in red in Figure 11b. Similar to 2000–2007, the rapid conversion of shrimp-farming area from other land cover continued. In 2007–2013, besides cultivated land (1595 hectares, 20.23%), a considerable amount of bare land (3252 hectares, 21.30%) was also converted to a shrimp farm, represented in sky blue and red, respectively (see Figure 11c). At the same time, 431 hectares (11.99%) of shrimp-farming area were transferred to bare land, represented in red in Figure 11d. In 2013–2018, 487 hectares (10.73%) of water bodies and 138 hectares (1.29%) of bare land were converted to a shrimp farm, represented in dark blue and red in Figure 12a. In addition, a significant amount of shrimp-farming area was transferred to cultivated land and bare land, a total of 2110 hectares (29.65%) and 1653 hectares (23.22%), respectively, represented in sky blue and red in Figure 12b.

The result shows that overall in 2000–2018, 1569 hectares of cultivated land were converted to shrimp farm, accounting for 6.86%, represented in sky blue in Figure 12c. On the other hand, an unusual amount of shrimp-farming area was transferred to bare land, cultivated land, and vegetation area, accounting for 564 hectares (25.11%), 383 hectares (17.01%), and 392 hectares (17.47%), respectively (see Figure 12d). Figure 13 is a vector layer sketch map showing overall gain, loss, and no change of shrimp-farming area from 2000–2018.

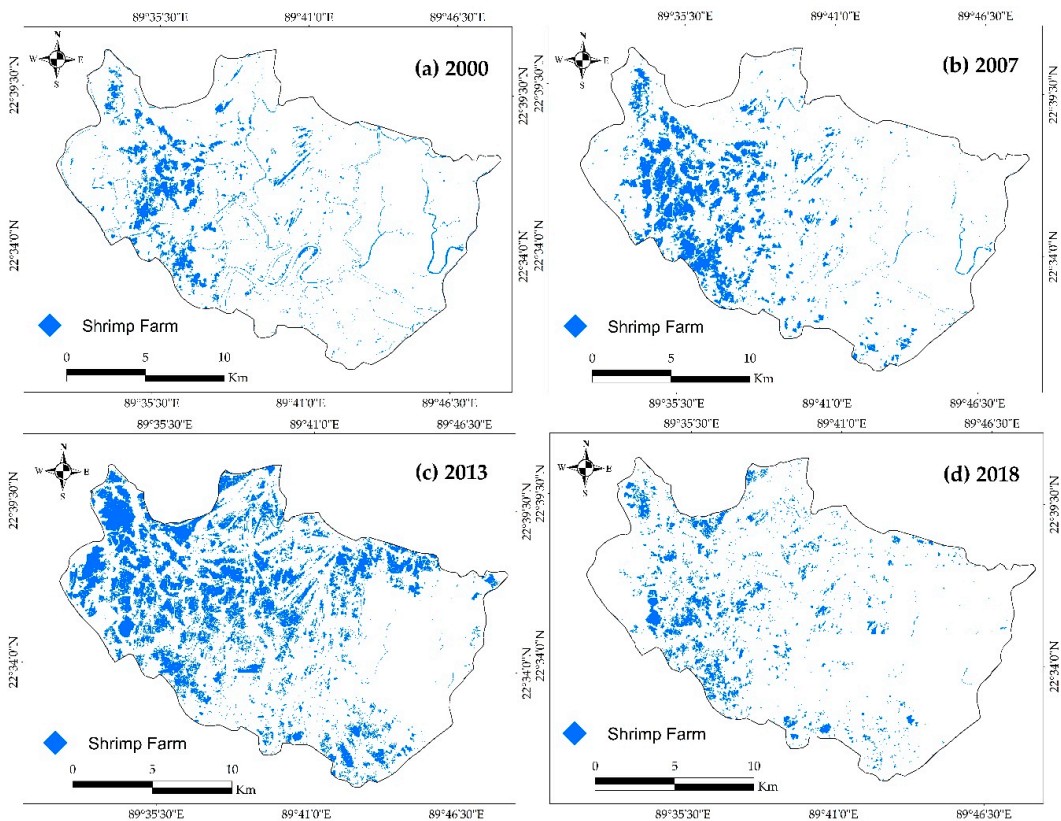

**Figure 10.** Changes in shrimp-farming area, Rampal, 2000–2018: (**a**) 2000; (**b**) 2007; (**c**) 2013; and (**d**) 2018.

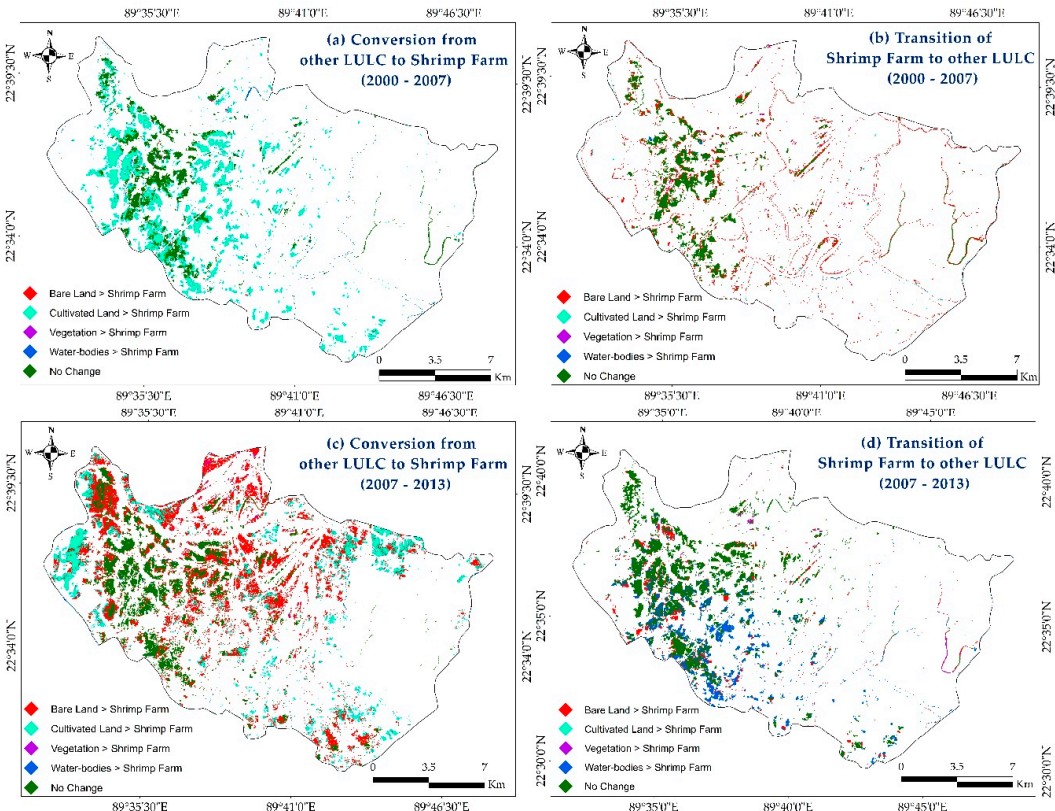

**Figure 11.** Conversion of shrimp-farming area, Rampal, 2000–2007 and 2007–2013.

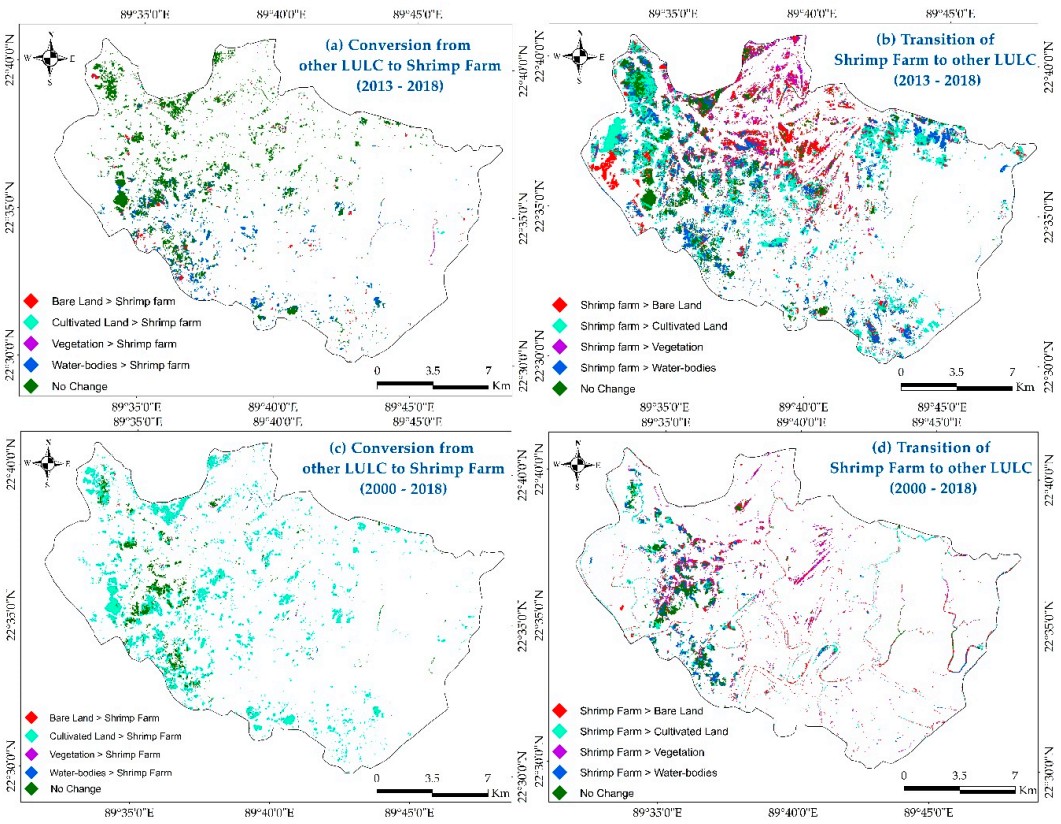

**Figure 12.** Conversion of shrimp-farming areas, Rampal, 2013–2018 and 2000–2018.

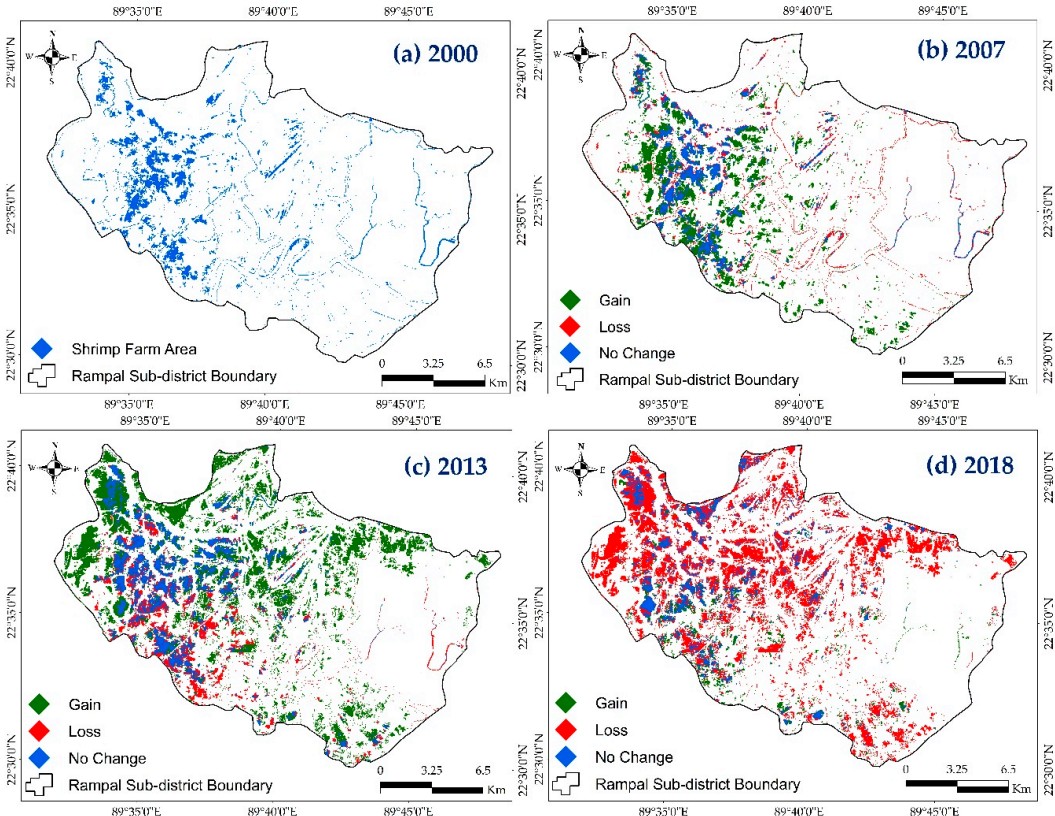

**Figure 13.** Spatiotemporal changes of shrimp-farming area, Rampal, 2000–2018. Green, red, and blue represent gain, loss, and no change, respectively.

3.3.2. Introduction of Rampal 1320 MW Coal-Fired Thermal Power Plant and Decline in Shrimp Farming

The introduction of the coal-fired thermal power plant and the decline in shrimp-farming area and yield in Rampal are connected. According to Figure 14a, in 2000, most of the power plant site and its surrounding area was used as cultivated land and a small amount of shrimp-farming area.

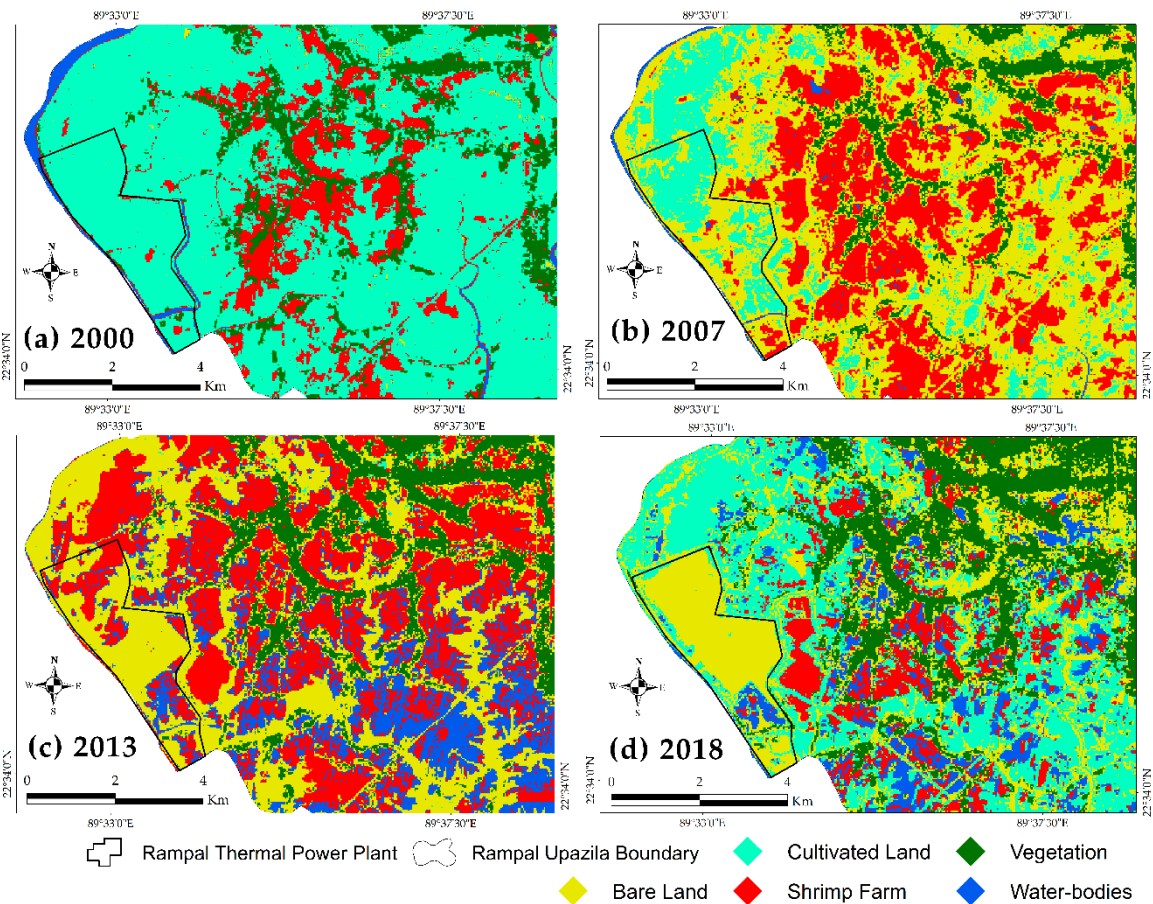

**Figure 14.** Spatiotemporal LULC of Rampal thermal power plant and surrounding area, 2000–2018: (**a**) 2000; (**b**) 2007; (**c**) 2013; and (**d**) 2018.

In 2007, farmers in this area started to convert their cultivated land into shrimp farms, as evidenced by Figure 14b. In 2013, the popularity of shrimp farming reached its highest level, as shown by the majority of the land of northwestern Rampal along with the power plant project site used for shrimp farming (see Figure 14c). However, Figure 14d confirms that after construction work on the thermal power plant began in 2017, the site and the surrounding area were gradually transferred from shrimp farm to bare land and cultivated land. According to South Asia for Human Rights (SAHR) (2015), approximately 400 acres of land, including a natural canal, were landfilled due to preparatory construction work [35], as shown in Figure 14.

## 4. Discussion

### 4.1. Shrimp Yield Is Declining in Bagerhat District and Increasing in Khulna and Satkhira Districts

The brisk and boundless expansion of bare land, influenced by the construction of the coal-fired thermal power plant in late 2013, seems to be one of the primary causes of the instability of the shrimp yield and farming area of Rampal. In addition, the impact of recent frequent climatic variable

changes, the outbreak of disease at shrimp farms, high labor costs, low shrimp prices, and long-term environmental consequences are equally responsible for the declining shrimp yield and farming area.

Our results show, however, that from 2010 to 2014, the shrimp yield of the southwestern coastal districts increased significantly compared to 2002–2009. The hectare-wise shrimp yield of the three coastal districts subsequently increased in 2015–2017. The overall shrimp production in Bangladesh has been increasing on a per-hectare basis and in the industry as a whole. The hectare-wise shrimp yield more than doubled in 2017 compared to 2002. This is valid for all three coastal districts in this research. However, in Bagerhat, shrimp yield (per hectare) has decreased compared to the other two districts in recent years, although in general, it was still higher compared to Satkhira and Khulna districts from 2002 to 2014.

The shrimp yield of Bagerhat district is associated with both increased and decreased hectare-wise shrimp production and shrimp-farming area. So, the decrease of one variable will affect the overall yield. Our study suggests that the brisk increase of bare land, led by the construction work of the thermal power plant, appears to be a critical factor behind the declining shrimp yield and farming area in Rampal. Apart from the power plant site, previous studies have suggested that there are some other factors equally responsible for the decline in shrimp farming. In recent years, the prices of shrimp have been low, but the labor cost remains high [22], so farmers are not stocking many farms/ponds. The frequent outbreak of disease at shrimp farms is another critical factor [24]. Besides, the long-term environmental consequences of shrimp farming, such as the loss of biodiversity and increased salinity [76], are equally accountable for the decrease of shrimp farming. Climatic variables such as cyclones [22,76], coastal flooding [77], drought [78], sea-level rise [1], and sea-level temperature [79] are severe threats to the growth of shrimp as well as the development of the shrimp sector in the coastal district.

*4.2. More than 70% of the Shrimp-Farming Area in Rampal Was Lost since the Introduction of the Coal-Fired Thermal Power Plant*

Rampal, in Bagerhat district, is important for long-term national food security, because it is one of the principal shrimp-producing subdistricts of Bangladesh. Therefore, Rampal was chosen as the study area in order to find the influencing factors behind the declining shrimp-farming area and shrimp yield.

However, the documented trend reveals that the shrimp-farming area of Rampal gradually increased during 2000–2013 and declined rapidly in 2018. There are many critical factors behind the declining shrimp-farming area, and the yield has been identified in previous studies. However, the increase of bare land, especially after construction work on the coal-fired thermal power plant began, seems to be a critical factor behind the declining shrimp-farming area in recent years.

The government acquired a total of 742.51 hectares (1834 acres) of land for the construction of two adjoining power plants [25] and only 34.82 hectares (86 acres) were Kash (state-owned) land; the rest was public land, which was used for rice and fish cultivation [26]. The government promises a massive boost to the power production of the country; however, there are concerns that it will have numerous irrecoverable adverse effects on the Sundarbans and the surrounding land and aquaculture, including shrimp farming [36]. Islam and Tabeta (2016) and other researchers suggest that it will trigger severe ecological damage during and after the construction of the plant due to forest clearing, leaked oil, waste disposal, and so forth [40,80,81]. At the land acquisition phase of the power plant, so many shrimp farmers were forced to stop farming and give away their land in and around the power plant site. The location of the power plant is close to the seaport, EPZ, and the proposed airport, so many industries acquired land nearby so that they can minimize the cost of transporting their goods and foster their business. The present study reveals that the area where the power plant is being constructed was previously a shrimp-farming area, and now the power plant and its surrounding area have been converted to bare land and cultivated land. Although various reports have suggested that thermal power plants have been the leading cause of the decline in shrimp farming in Rampal [35–37,39,40,80,82], an analysis using satellite imagery estimates that approximately

4989.33 hectares (50 km$^2$), or nearly 70% of the shrimp-farming area has been lost in Rampal since December 2013, and all the factors discussed are equally responsible. If the current pattern of decline persists, we fear that the shrimp-farming area will soon cease to exist. The data and results from this study can be used to detect changes in shrimp farming in the future with higher intensity. This research might help to plan more stable shrimp-farming practices as well through conservation and sustainable aquaculture practices.

## 5. Conclusions

The livelihoods of the three significant shrimp-farming coastal districts of Bangladesh, Bagerhat, Satkhira, and Khulna, largely depend on shrimp-farming activities. However, the shrimp yield and shrimp-farming area of Bagerhat district has decreased compared to Satkhira and Khulna districts in recent years. The introduction of the Rampal thermal power plant appears to be a critical factor behind the declining shrimp-farming area and shrimp yield of Rampal, Bagerhat district, in recent years. The government of Bangladesh acquired the power plant site over the shrimp-farming area in order to boost the power production of the country. Climatic variables, natural and environmental consequences, disease outbreaks, low shrimp prices, and high labor cost are some of the other notable factors that seem to account for the declining shrimp-farming area and yield of Bagerhat district. This research revealed that over 70% of the shrimp-farming area was lost in Rampal since December 2013. In this research, based on the shrimp yield dataset (SYD) and k-means classification, we quantified the differences in shrimp yield of three southwestern coastal districts between 2002 and 2017. In addition, we generated temporal shrimp-farming area change maps and identified the influencing factors behind the declining shrimp-farming area in Rampal between 2000 and 2018 based on satellite imagery and maximum likelihood (ML) classification. Different researchers have applied land cover change analysis based on satellite images to different parts of Bangladesh in the past few years; the present study introduces a new tool for monitoring the evaluation of aspects of the aquaculture industry and land use planning. It is high time that the government declare the Rampal subdistrict, along with other primary shrimp-farming coastal subdistricts, a "shrimp zone". Moreover, the government should implement an effective policy to protect the vulnerable shrimp-farming industry and shrimp farmers in the southwestern coastal districts of Bangladesh to fulfill the sustainable development goal, considering it is the second-largest export earning source of the country after ready-made garments. This kind of research has good potential to compensate for sustainable development, from the local to the global level, all over the world. Furthermore, the results of this research could be useful to policymakers, planners, and other researchers who are interested in utilizing these solutions for different studies.

**Author Contributions:** M.F.K. conceptualized the overall ideas of this study. M.F.K., X.Z., and R.L. designed the research. M.F.K. and R.L. performed the data analysis. M.F.K. wrote the manuscript, and all authors were involved in improving the quality of this manuscript. X.Z. collected funds, was the project administrator, and supervised the research.

**Funding:** This research was funded by the National Natural Science Foundation of China, grant number [41671384; 41301410], and the National Key Research and Development Program of China grant number [2017YFB0503500].

**Acknowledgments:** This research work is part of a graduate paper for the Master of Science in Cartography and Geographical Information Engineering program at the School of Resource and Environmental Science, Wuhan University. The author's thanks the two reviewers, as well as Akib Javed and Patrik Silva for their valuable suggestion.

**Conflicts of Interest:** The authors declare no conflict of interest.

## Appendix A

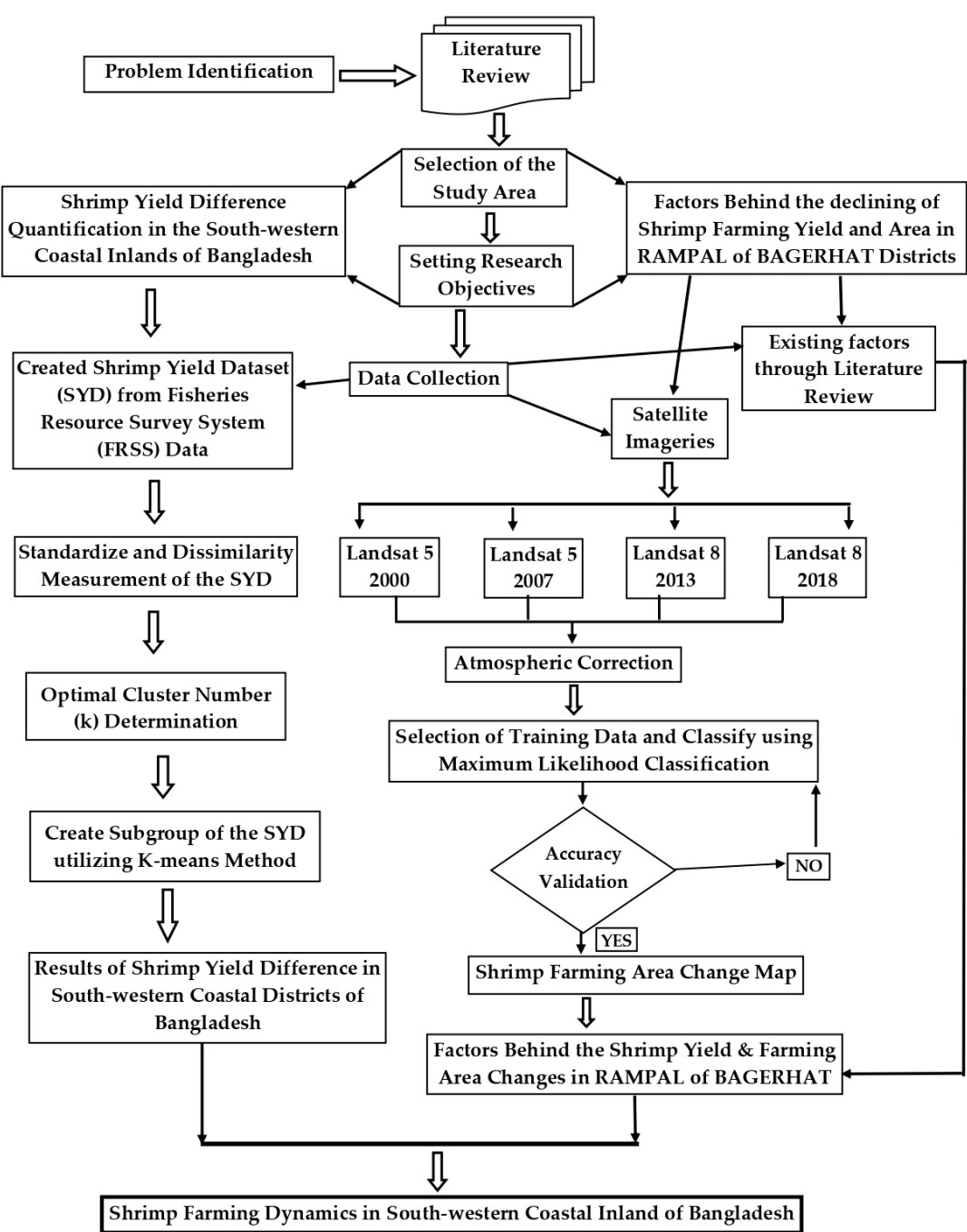

**Figure A1.** Sequential explanation (conceptual framework) of the research process.

**Table A1.** Spatiotemporal transition of LULC in Rampal, 2000–2007 (in hectares).

| | | 2000 | | | | | |
|------|------------|------------|------------------|-----------------|--------------|------------------|----------------|
| | **LULC Class** | **Vegetation** | **Water Bodies** | **Shrimp Farms** | **Bare Land** | **Cultivated Land** | **Class Total** |
| **2007** | Vegetation | 4614 | 0 | 46 | 23 | 477 | 5160 |
| | Water bodies | 0 | 643 | 37 | 1 | 030 | 712 |
| | Shrimp farms | 25 | 47 | 1283 | 18 | 2220 | 3593 |
| | Bare land | 1247 | 402 | 861 | 336 | 12,421 | 15,267 |
| | Cultivated land | 7 | 13 | 21 | 106 | 7739 | 7886 |
| | Class total | 5893 | 1106 | 2247 | 484 | 22,888 | 32,618 |

**Table A2.** Spatiotemporal transition of LULC, 2007–2013 (in hectares).

| | | 2007 | | | | | |
|------|------------|------------|------------------|-----------------|--------------|------------------|----------------|
| | **LULC Class** | **Vegetation** | **Water Bodies** | **Shrimp Farms** | **Bare Land** | **Cultivated Land** | **Class Total** |
| **2013** | Vegetation | 4756 | 0 | 49 | 1946 | 63 | 6814 |
| | Water bodies | 3 | 403 | 934 | 2252 | 951 | 4543 |
| | Shrimp farms | 43 | 50 | 2178 | 3252 | 1595 | 7118 |
| | Bare land | 348 | 258 | 431 | 7049 | 2618 | 10,704 |
| | Cultivated land | 10 | 0 | 2 | 768 | 2659 | 3439 |
| | Class total | 5160 | 711 | 3594 | 15,267 | 7886 | 32,618 |

**Table A3.** Spatiotemporal transition of LULC in Rampal, 2013–2018 (in hectares).

| | | 2013 | | | | | |
|------|------------|------------|------------------|-----------------|--------------|------------------|----------------|
| | **LULC Class** | **Vegetation** | **Water Bodies** | **Shrimp Farms** | **Bare Land** | **Cultivated Land** | **Class Total** |
| **2018** | Vegetation | 6085 | 150 | 635 | 1687 | 109 | 8666 |
| | Water bodies | 4 | 1203 | 1249 | 280 | 27 | 2763 |
| | Shrimp farms | 21 | 487 | 1471 | 138 | 10 | 2128 |
| | Bare land | 683 | 991 | 1653 | 4732 | 833 | 8892 |
| | Cultivated land | 21 | 1711 | 2110 | 3868 | 2459 | 10,168 |
| | Class total | 6814 | 4542 | 7118 | 10,705 | 3438 | 32,617 |

**Table A4.** Spatiotemporal transition of LULC in Rampal, 2000–2018 (in hectares).

| | | 2000 | | | | | |
|------|------------|------------|------------------|-----------------|--------------|------------------|----------------|
| | **LULC Class** | **Vegetation** | **Water Bodies** | **Shrimp Farms** | **Bare Land** | **Cultivated Land** | **Class Total** |
| **2018** | Vegetation | 5129 | 30 | 392 | 102 | 3013 | 8666 |
| | Water bodies | 4 | 377 | 387 | 16 | 1978 | 2763 |
| | Shrimp farms | 10 | 11 | 520 | 19 | 1569 | 2128 |
| | Bare land | 710 | 343 | 564 | 225 | 7050 | 8892 |
| | Cultivated land | 41 | 345 | 383 | 122 | 9277 | 10,168 |
| | Class total | 5894 | 1106 | 2246 | 484 | 22,887 | 32,617 |

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
