# Peer review of "Dynamics of Shrimp Farming in the Southwestern Coastal Districts of Bangladesh Using a Shrimp Yield Dataset (SYD) and Landsat Satellite Archives"

_sustainability, doi:10.3390/su11174635_

Round 1
Reviewer 1 Report
The work itself is good. However, the manuscript is too long with unnecessarily details on the methods and so on and needs to shortened quite a bit. Extensive English correction is required. I would love to see the shorter version of this paper with high resolution figures. Here are just some examples:
Abstract
Abstract is too long. Please try to shorten it especially the results part
Introduction
Line 53: Akber et al (2017) …not Mr. Akber
Line 59-61: Rearrange the sentence
Line 62- 64: The common supervised and unsupervised clustering methods for change analysis are…? (please rearrange the sentence)
Line 66-75: These are the literature review for K-means cluster analysis. Please re-arrange all the lines. Start the sentence like.. Agarwal et al (2014) used k means clustering to study….. and so on.
Line 77: small letter ‘the’
Linev100-102: Re-arrange the sentence
Line 108-110: The common supervised land cover classification methods are….??
Line 116: Delete the sentence “all-round expertise” and replace it with what you actually did
Line 118: Delete “tremendous significance”. Please do not use these kind of heavy terms
Line 125: Furthermore
Materials and Methods
Line 141: Inset not Insert
Figure 1: Increase the resolution of the figure. The green area in Landsat image doesn’t seem to be correct, please explain. You do not need to put the projection, scale factor etc. information in this figure. Within the figure, please separate each one like a, b, and c and briefly explain them.
Try to shorten the description of your study area
Figure 2: Inset map not Insert. Both the right and left map cannot be inset map. Also work on the resolution of this map.
Line 84-85: Please re-arrange the sentence.
Figure 3: You need to have several training areas for each land cover type in order to do image classification. Please explain.
Line 294-297: You do not need this.
Try to shorten this section.
Results:
Too many figures. Figures could be merged. Some figures (e.g. 11, 12, 15) very hard to read. Please try to get the resolution better. Try to shorten your results section. Avoid unnecessarily explanations. Figures 13 and 14 could be mentioned in the text??
Reviewer 2 Report
The paper presents some interesting new tools for evaluation of aspects of the aquaculture industry and land use planning.
1. The English needs considerable corrections. In many cases verbs are missing, tenses do not match, a few misspellings, and in several locations run on sentences that need to be broken into smaller sentences that will make the reading more smooth. Mixing up districts and regions within a district is confusing to the reader. Need to be more clear and avoid double counting.
2. There is one glaring problem with the paper that needs to be addressed in a revision. There is confusion by the authors with yield per hectare and number of hectares. There seems to be an attitude that the industry is regressing as hectares have been converted to other uses. But the production of shrimp in Bangladesh has been increasing on a per hectare basis as well as an industry as a whole. There are also many reasons why ponds are not being farmed. Prices have been low, so farmers are not stocking as many ponds, hatchery disease and production issues have limited the number of post larvae that could be stocked, so ponds may not be prepared and stocked. With greater intensity of farming farmers can produce more shrimp from fewer hectares.
3. Correlation is not necessarily related to causation. The fact that there may be fewer hectares being farmed may be correlated with the reduction of ponds being stocked each year. But there are many reasons for this and the discussion totally misses the many reasons, but instead putting all blame on the coal fired thermo-plant.
4. After several trips to these areas and visiting farms and processing plants, I have never heard anyone refer to the area as the “Kuwait of Bangladesh”. It also makes little sense as oil from Kuwait is a natural resource owned by the government and farmed shrimp are privately owned farmed animals.
Round 2
Reviewer 1 Report
Line 137: no need 'in this paper'
Reviewer 2 Report
Much improved revision. Authors were diligent to address comments and edits by the two reviewers. A few minor formatting issues (word wrap) and margins, should be checked and fixed before printing.